# Wear Behavior of Ductile Iron Wheel Material Used for Rail-Transit Vehicles under Dry Sliding Conditions

**DOI:** 10.3390/ma13122683

**Published:** 2020-06-12

**Authors:** Lifeng Tong, Qingchuan Zou, Jinchuan Jie, Tingju Li, Zhixin Wang

**Affiliations:** 1Key Laboratory of Solidification Control and Digital Preparation Technology, School of Material Science and Engineering, Dalian University of Technology, Dalian 116000, China; jiejc@dlut.edu.cn (J.J.); tjuli@dlut.edu.cn (T.L.); zxwang@dlut.edu.cn (Z.W.); 2Key Laboratory for Ecological Metallurgy of Multimetallic Mineral of Ministry of Education, School of Metallurgy, Northeastern University, Shenyang 110819, China; zouqingchuan@mail.neu.edu.cn

**Keywords:** ductile iron, rail-transit wheel, heat treatment, wear resistance, thermal property

## Abstract

A ductile iron wheel used for a rail-transit vehicle was treated with a recommended heat-treatment process. The ductile iron wheel after heat treatment was composed of graphite nodules and tempered sorbite with an area fraction of 98%. A friction test of the ductile iron and carbon steel wheel materials was systematically performed under different normal loads and sliding velocities. The results indicated that the wear mechanism of the ductile iron wheel changed from adhesion to abrasion with an increase in the normal load level. Adhesion was the main wear mechanism at different sliding velocities and normal load level. The impact of the normal load on the wear mechanism was greater than that of the sliding velocity. Since the ductile iron wheel material had excellent thermal property and higher carbon content, it exhibited a lower wear rate, a smaller difference value of the friction coefficient, and plastic deformation on the worn surface than those of the carbon steel wheel material. This indicates that ductile iron wheels may have a longer wear life, greater traction, and higher stability during operation than carbon steel wheels. The iron wheels have the potential for being applied in rail-transit vehicles.

## 1. Introduction

Rail-transit vehicles are usually equipped with rolled steel or carbon cast steel wheels. However, unavoidable defects in steel wheels affect their application [1,2,3,4,5,6]. A large amount of friction heat is generated during the braking process due to the low thermal conductivity of the steel wheel. This leads to welding between the rim and brake rotor, causing the vehicle to shut down. A directional fiber crystal structure is produced in rolled steel wheels during the rolling process. The brake rotor and wheel rub against each other during the braking process, and the material is delaminated from the steel rolled wheel surface, which accelerates the degradation of the rolled steel wheel. Considerable mass loss occurs due to the wear and tear of the steel-cast wheel. The roundness of the wheel decreases after a long service process, which causes the vehicle to vibrate and generate noise. Crack accidents of the rolled steel wheel rim may occur during the running process, which is a serious safety hazard for train operation. These factors hinder the application of steel wheels in rail-transit vehicles. 

Hervas I., Murcia S.C., and Kruthiventi S. et al. [7,8,9,10,11,12,13,14,15] reported extensively on the industrial application of ductile iron due to its excellent casting performance, machinability, wear resistance, shock absorption, high flaw tolerance, good hardenability, high thermal conductivity, low thermal expansion coefficient, and low cost. Therefore, it is feasible to use ductile iron to produce rail vehicle wheels. Researchers have investigated the production of ductile iron wheels and examined their mechanical properties. J. Li et al. [16] produced ductile iron train wheels that were heat-treated via austempering. The mechanical properties were 790 MPa in tensile strength, 13.0% in elongation, and 238 HB in Brinell hardness. 

The wear resistance of ductile iron has also been studied in recent years. H.R. Abedi et al. [17] studied the effect of the nodule count on the sliding wear behavior of a ferritic-pearlitic ductile iron and found that the specimens with the high nodule count exhibit a lower wear rate than those having the low nodule count at the lower applied loads, while wear resistance deteriorates with increasing nodule count at the higher loads. O. Celik et al. [18] examined high temperature abrasive wear behavior of an as-cast ductile iron at a temperature range between 50 and 100 °C. The highest resistance to abrasive wear was observed at a temperature range between 50 and 100 °C. At this temperature range, the ductile iron exhibited more than 15% higher abrasion resistance than room temperature. Y. Sahin et al. [19] investigated the effect of the martensite volume fraction (MVF) and tempering time on the abrasive wear of ferritic ductile iron. The results showed that weight loss resistance and strength increased and ductility decreased with increasing MVF. B. Podgornik et al. [20] introduced a new method, which, through local reinforcement with inserts, improved wear resistance of ductile iron without compromising other properties. Results of the present investigation showed that tribological properties of ductile iron could be greatly improved by local surface reinforcement. Through the formation of carbides, a hard transition or functional gradient zone was formed around the inserts, which then carried the load and improved wear resistance of ductile iron. At the same time, it maintained low friction.

Although some studies have been devoted to producing ductile iron train wheels and improving their performance, little attention has been paid to their friction and wear performance. The present study mainly focused on the wear behavior of a ductile iron train wheel under different loads at different sliding velocities under dry sliding conditions. Test results were compared with those for a carbon steel wheel, and the corresponding mechanisms were discussed.

## 2. Experimental

As-cast ductile iron was selected as the testing material for producing a rail-transit vehicle wheel work blank, which was cast through the following route. Molten iron with a temperature range of 1470–1490 °C was poured into a tundish where the nodularizer and the inoculant were already placed. The nodularizing treatment occurred between molten iron and the nodularizer. The molten iron after nodularizing treatment was poured into the sand mold to form the wheel work blank. Then, the wheel work blank was roughly machined to the size of a railway vehicle wheel. The ductile iron wheel was heat-treated in a resistance furnace. The wheel was austenitized at 900 °C (1173.15 K) for 240 min and quenched in a 15%–17% UCONA (A quenchant, which possesses several advantages including non-toxic, non-corrosion, smokeless and odorless. Its cooling rate is adjustable between that of water and oil.) aqueous solution for >20 min. Subsequently, the ductile iron wheel was tempered at 570 °C (843.15 K) for 180 min and cooled to <200 °C (473.15 K), which was followed by air cooling. The schematic process of heat treatment for the ductile iron wheel is shown in Figure 1.

The microstructure of testing materials was observed by an optical microscopy (OM, GX53, Olympus, Tokyo, Japan). The phase content of testing materials was measured according to ‘ASTM A668/A668M-2014 Standard Specification for Steel Forgings, Carbon and Alloy, for General Industrial Use.’ The nodularity and graphite nodule size were defined according to the ‘ASTM A247-17 Standard Test Method for Evaluating the Microstructure of Graphite in Iron Castings.’

A carbon steel wheel was used for comparison. The carbon steel wheel was austenitized at 900 °C (1173.15 K) for 240 min and cooled in air. Subsequently, the carbon steel wheel was tempered at 580–600 °C (853.15–873.15 K) for 240 min and cooled to <200 °C (473.15 K), which was followed by air cooling. Performance comparisons were conducted among the carbon steel, as-cast, and heat-treated ductile iron wheels. The testing materials were taken from the wheel rim. The hardness was tested using a Brinell hardness tester (THBS-3000E, Time, Beijing, China) with a pressuring ball diameter of ∅10 mm, a loading force of 3000 N, and a dwell time of 15 s. The tensile tests were examined in an electronic universal tester (DNS100, WTS, Jinan, China) using a tensile specimen with a diameter of 10 mm and an original gauge length of 50 mm. The impact tests were carried out in a Charpy impact machine tester (JB-300B, Time, Jinan, China) using an impact specimen with a U notch. The thermal properties, including the thermal diffusivity, specific heat, and thermal conductivity of the two wheel materials were determined using a thermal-property analyzer (Flashline^TM^-5000, Anter, Pittsburgh, PA, USA). The microstructural features were examined using a metallographic microscope (OM, GX53, Olympus, Tokyo, Japan). The carbon and sulfur contents in the testing materials were determined using a carbon-sulfur analyzer (CS-8800, Jinyibo, Wuxi, China), and other compositions were determined using a fluorescence spectromter (XRF-1800, Shimadzu, Japan). Friction and wear tests were performed using a block-on-ring material surface performance comprehensive tester (CFT-1, Zkkh, Lanzhou, China) under dry sliding. Friction and wear tests were conducted under two conditions. The first condition included: (i) normal loads of 50, 60, and 70 N, respectively, a sliding velocity of 6.3 × 10^4^ mm/min, and a duration of 30 min. The second condition included: (ii) a normal load of 50 N, sliding velocities of 6.3 × 10^4^, 9.42 × 10^4^, and 12.54 × 10^4^ mm/min, and durations of 30, 20, and 15 min, respectively. The comparison wear tests of a ductile iron wheel and a carbon steel wheel were conducted under the same test conditions and the friction length of each sample under two conditions was guaranteed to be the same during the friction test to ensure an effective comparison among samples. Ductile iron and carbon steel samples were 19 mm in length and 12 mm in width and thickness. The friction pair, which was taken from the rail, was ∅40 mm in diameter and 10-mm thick. The size of the wear samples and configuration of the wear tester are shown in Figure 2. A certain force, which was mediated by the transducer, was applied to the sample. The motor drove the axis to rotate the friction pair. The friction pair rubbed on the surface of the sample under a certain normal load at a certain sliding velocity. The friction coefficients and curves were recorded using a computer connected to a wear tester. The average value of the friction coefficient was adopted in this study. A thermocouple was placed below the worn surface of the sample. The temperature of the worn surface was recorded using a temperature recorder connected to the thermocouple during the friction process. The samples were weighed before and after the friction and wear tests to measure the mass loss. The block and counterpart surfaces were cleaned with acetone. The morphology and composition of the worn surface of the specimens were analyzed using a scanning electron microscope (Zeiss Supra 55, Zeiss, Oberkochen, Germany) equipped with an energy-dispersive X-ray spectrometer and a laser scanning confocal microscope. An image analysis software OLYCIA m3 (BH-RJ007, Bahens, Shanghai, China) was used to analyze the phase content of testing materials and the morphology of a graphite nodule. The micrographs (×100) of the ductile iron sample without etching (preventing the interference of pearlite) were taken using a metallographic microscope. The micrographs were analyzed by the graphite analysis function of OLYCIA m3. A report including nodularity, graphite nodule size, and area fraction of graphite was obtained. The micrographs of samples after etching were taken again. A report including the area fraction of phases and graphite nodules was obtained. The phase content of testing materials was obtained after calculation. Three tests were performed for each specimen condition, and the average values are reported herein.

## 3. Results and Discussion

### 3.1. Microstructure and Mechanical Property 

The ductile iron wheel is shown in Figure 3a. Metallographic specimens obtained from the wheel rim were observed. The as-cast microstructure analysis revealed nodularity of 89% and a graphite nodule diameter range of ∅30–60 μm. The microstructure was composed of ferrite and pearlite with an area fraction of 38.7% as well as graphite nodules, which is shown in Figure 3b.

The microstructures of the testing materials are shown in Figure 4. The ductile iron wheel after heat treatment was composed of graphite nodules, retained austenite, and tempered sorbite with an area fraction of 98% (see Figure 4a). The microstructure of the carbon steel wheel comprised ferrite and pearlite with an area fraction of 81% (see Figure 4b). The counterpart, which is known as the steel rail, consisted of ferrite and pearlite with an area fraction of 95% (see Figure 4c). Table 1 presents the chemical compositions of the cast ductile iron wheel, carbon steel wheel, and rail, which are labeled as ‘D,’ ‘S,’ and ‘R,’ respectively.

The mechanical properties of the testing materials are presented in Table 2. The tensile strength of the ductile iron wheel material after heat treatment (D’) was significantly improved when compared to that of the as-cast ductile iron wheel (D) and was close to that of the carbon steel wheel. The ductile iron wheel had a higher yield strength than the carbon steel wheel, which indicated that it had a greater resistance to elastic deformation. The hardness value of the ductile iron wheel increased with the heat treatment and became close to that of the carbon steel wheel. The elongation and impact energy of the ductile iron wheel were lower than those of the carbon steel wheel, which suggested that the ductile iron wheel was inferior to the carbon steel wheel in some aspects.

### 3.2. Thermal Property

The thermal properties of materials affect their wear resistance. Excellent thermal conductivity allows the dissipation of heat, which reduces the mass loss. The thermal diffusivity, specific heat, and thermal conductivity of the ductile iron wheel after heat treatment (D’) and the carbon steel wheel (S) are presented in Table 3. These three parameters were higher for the ductile iron wheel than for the carbon steel wheel, which indicated that the ductile iron wheel had better thermal properties. This helped the ductile iron wheel to dissipate heat faster during the operation.

The temperatures of the ductile iron and carbon steel wheels in the process of friction under conditions (i) and (ii) are shown in Figure 5. The friction temperature of both wheels increased with load and sliding velocity. The temperature of the ductile iron wheel was lower than that of the carbon steel wheel under the same condition. The ductile iron wheel dispersed heat faster than the carbon steel wheel due to the ductile iron wheel having better thermal properties. 

### 3.3. Worn-Surface Analysis

The roughness, wear depth, and grinding crack width of the ductile iron and carbon steel samples under conditions (i) and (ii) are listed in Table 4 and Table 5, respectively. The roughness used in this paper is an arithmetical mean deviation of the profile of the worn surface (Ra). It is a roughness evaluation parameter based on regional topography of the detected area. The worn surface roughness was measured in a direction perpendicular to the friction direction of the friction ring. As indicated by Table 4, the wear depth and grinding crack width exhibited an increasing trend with an increase in the normal load. According to the formula $F=\mu \times F_{N}$, the friction *F* increases with the normal load *F_N_*, which raises the wear rate. Hence, the wear depth and grinding crack width increased. As indicated by Table 5, the ductile iron wheel exhibited an increase and then a decrease in the wear depth and grinding crack width with the increasing sliding velocity under 50 N. The foregoing results indicate that the wear depth and grinding crack width were consistent with the wear rate. Additionally, the wear depth and grinding crack width of the ductile iron wheel were significantly smaller than those of the carbon steel wheel under the same conditions (see Table 4 and Table 5). These results also indicate that the ductile iron wheel had better wear resistance than the carbon steel wheel.

Figure 6 shows scanning electron microscopy (SEM) images of the worn surfaces of the ductile iron and carbon steel wheel specimens under condition (i). The energy-dispersive X-ray spectroscopy (EDS) results of the view field in Figure 6 are presented in Table 6. For the worn surface under a low normal load (50 N), the adhesion was a dominant wear mechanism of both specimens (see Figure 6a,b). Under a medium normal load (60 N), furrow and peeling appeared on a worn surface of a ductile iron specimen (see Figure 6c). The ductile iron specimen surface was subjected to a tangential stress by a unidirectional cyclic friction of the friction ring. A material flake forms on the worn surface [17], which is called peeling. The flake might break off of the matrix due to the low ductility or may still be adhered to on the worn surface [21]. The adhesion was still the main wear mechanism on the worn surfaces of the carbon steel specimen (see Figure 6d). Under a high normal load (70 N), plentiful furrows were formed along the sliding direction and particles could be observed on the worn surface of both specimens (see Figure 6e,f). It was considered that depth of scratches increased when the abrasive particles reached the leading edge of the graphite nodule [15]. Therefore, the furrows on the worn surface of a ductile iron specimen were more clear than that of the carbon steel specimen. Abrasion was the main wear mechanism under this condition. Oxidation occurred throughout the wear process.

Figure 7 shows SEM micrographs for the worn surfaces of the ductile iron and carbon steel wheel specimens under condition (ii). The energy-dispersive X-ray spectroscopy (EDS) results of the view field in Figure 7 are exhibited in Table 7. At a low sliding velocity (6.3 × 10^4^ mm/min), adhesion was observed on the worn surface of both specimens (see Figure 7a,b). At a medium sliding velocity (9.42 × 10^4^ mm/min), oxidation and adhesion were the main wear mechanisms on the worn surface of two specimens (see Figure 7c,d). Some furrows appeared on the worn surface of a carbon steel specimen. At a high sliding velocity (12.54 × 10^4^ mm/min), adhesion aggravated on the worn surface of both specimens (see Figure 7e,f). As can be seen from the above, adhesion dominated the wear process of both specimens at various sliding velocities. The higher sliding velocity produced a large amount of heat, which softened the worn surface. Furthermore, plastic deformation occurred on the worn surface during adhesion. These factors led to the lowest roughness (12.5 μm) and smallest friction coefficient (0.57) with the highest sliding velocity (12.54 × 10^4^ mm/min).

Under various experimental conditions, the wear mechanism of the ductile iron wheel mainly included adhesion and abrasion. The ductile iron possessed excellent thermal properties, which can dissipate friction heat faster than carbon steel. Thus, ductile iron wheels ran smoothly during the test. The thermal properties of the carbon steel wheel were worse than that of nodular iron wheels. The oxidation and softening degree of the carbon steel specimen was more serious due to the poor thermal property. This resulted in a significant increase in the mass loss of carbon steel wheels at a high sliding velocity.

### 3.4. Plastic-Deformation Behavior

Figure 8 and Figure 9 show the cross-sectional morphologies of the ductile iron and carbon steel wheels under conditions (i) and (ii), respectively. The worn surfaces are indicated by dotted lines. The arrow indicates the rolling direction of the friction ring. The plastic-deformation wear mechanism was characterized by significant surface deformation on the worn surface. The plastic-deformation was the change in microstructure under the worn surface of samples. The microstructure of the plastic-deformation area was different from the original microstructure after the test. Its direction was consistent with the rolling direction of the friction ring, which can be found in Figure 8 and Figure 9. The depth of plastic-deformation is the depth of this deformed microscopic region. For the ductile iron wheel material, the microstructure deformation was difficult to observe under a small normal load (see Figure 8a) or a low sliding velocity (see Figure 9a). Slight microstructure deformation was observed under higher loads (see Figure 8b,c) and sliding velocities (see Figure 9b,c). The ductile iron wheel possessed a similar increment in the plastic deformation under the increase of the normal load and the sliding velocity. The graphite nodules close to the worn surface were deformed to different degrees under the higher loads and sliding velocities (see Figure 8c and Figure 9c). For the carbon steel wheel materials, significant plastic-deformation under worn surfaces can be found even under a small load (see Figure 8d) or a low sliding velocity (see Figure 9d). The microstructure deformation became heavier gradually as the normal load (see Figure 8d,f) or sliding velocity (see Figure 9d,f) increased. The microstructure deformation of the worn surface was smaller for the ductile iron wheel specimens than that for the carbon steel wheel specimens under the same testing conditions. The nodular iron wheels possessed greater thermal properties than carbon steel wheels. The softening degree of nodular iron wheels was less than that of carbon steel wheels. The peeling and furrow produced less heat and had less effect on the plastic deformation of the worn surface than carbon steel wheels. Therefore, the nodular iron wheels possessed less plastic-deformation than carbon steel wheels.

### 3.5. Friction Coefficient 

The friction-coefficient curve provides important dynamic output information for the rolling contact friction system [17]. Figure 10 shows the friction-coefficient curves of the ductile iron and carbon steel wheels under conditions (i) and (ii). The results of the friction tests are presented in Table 4 and Table 5. The mechanical properties of graphite in the ductile iron are much lower than that of the matrix. Therefore, the graphite nodules could act as porosity defects even if they did not affect the roughness of the worm surface, which give ductile iron a higher friction coefficient than the carbon steel. The difference value of the friction coefficient is the difference between the maximum and minimum values in the curves. As indicated by Table 4 and Table 5, the difference value of the friction coefficient for the ductile iron wheel was smaller than that for the steel wheel under the same conditions in most cases. Due to poor thermal property, the adhesion and oxidation of carbon steel wheels was aggravated. The intermittent motion replaced the balanced motion between the carbon steel wheel and the friction pair, which might lead to greater friction vibration of carbon steel wheels than that of ductile iron wheels. This was the reason that the difference value of the friction coefficient of carbon steel wheels was greater than that of ductile cast iron wheels. Thus, the ductile iron wheel may afford greater traction and run more smoothly than the carbon steel wheel under the same load and sliding velocity.

### 3.6. Wear Rate

The wear rate is an important parameter of the wear resistance of a material, which represents the wear life of the material. The wear rates of both wheels under different conditions are presented in Figure 11 and Table 4 and Table 5. The wear rate of the ductile iron and carbon steel wheels increased under both test conditions. The wear rates of the ductile iron wheels changed relatively smoothly under both test conditions. The wear rate of the carbon steel wheels increased sharply with increase of the sliding velocity. An increasing sliding velocity could lead to a temperature increase, which reduces the hardness [20]. The increasing temperature of the contact surface also accelerates the formation of oxide. The wear rate is affected by the reduction of the hardness and the formation of oxide [22,23,24], which significantly increased the wear rate of the carbon steel wheels. Furthermore, the wear rate of the ductile iron wheel was significantly lower than that of the carbon steel wheel under the same conditions. The excellent thermal properties of the ductile iron wheel made the friction heat dissipate quickly, which improved its wear resistance. Furthermore, the ductile iron wheel had a higher carbon content than the carbon steel wheel. The study of Masaharu Ueda et al. [25] indicated that, if the carbon content of the material increases properly, the wear resistance also increases. The foregoing results suggest that the ductile iron wheel had better wear resistance than the carbon steel wheel.

## 4. Conclusions

A ductile iron wheel for a rail-transit vehicle was cast and treated with an appropriate heat-treatment process. A friction test revealed the wear behavior of the ductile iron wheel, and the results for ductile iron and carbon steel wheels were compared. The conclusions are as follows:

(1) The ductile iron wheel exhibited nodularity of 89% and a graphite diameter of ∅30–60 μm. After heat treatment, the ductile iron wheel comprised graphite nodules and tempered sorbite with an area fraction of 98%. The tensile strength of the ductile iron wheel after heat treatment was close to that of the carbon steel wheel. The ductile iron wheel had higher yield strength than the carbon steel wheel.

(2) With an increase in the normal load, the wear mechanism of the ductile iron wheel changed from adhesion to abrasion. With an increase in the sliding velocity, adhesion remained the main wear mechanism. The impact of the normal load on the wear mechanism was greater than that of the sliding velocity. Oxidation occurred throughout the wear process.

(3) Due to the excellent thermal properties and higher carbon content of ductile iron, the ductile iron wheel material exhibited a lower wear rate, a greater friction coefficient, and a smaller difference value of the friction coefficient and plastic deformation on the worn surface than those of the carbon steel wheel material.

## Figures and Tables

**Figure 1 materials-13-02683-f001:**
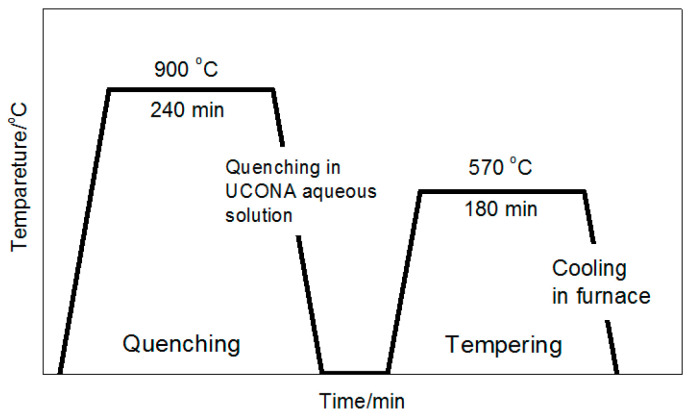
Schematic process of heat treatment for the ductile iron wheel.

**Figure 2 materials-13-02683-f002:**
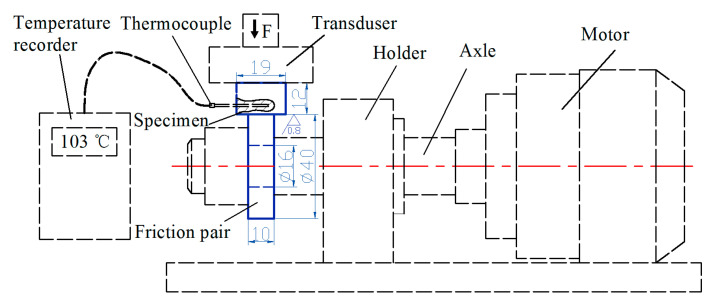
Size of the wear specimens and configuration of the wear tester.

**Figure 3 materials-13-02683-f003:**
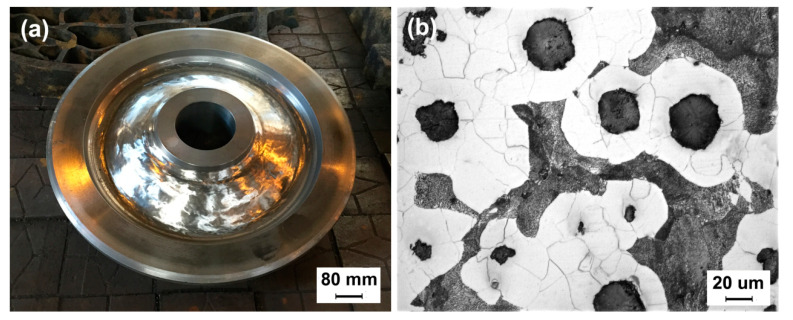
(**a**) Ductile iron wheel for rail-transit vehicles. (**b**) As-cast microstructure.

**Figure 4 materials-13-02683-f004:**
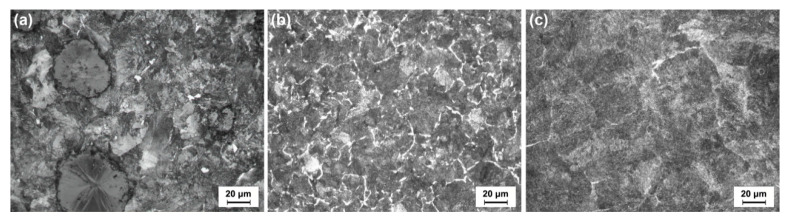
Microstructures of the testing materials: (**a**) ductile iron wheel after heat treatment, (**b**) carbon steel wheel, and (**c**) rail.

**Figure 5 materials-13-02683-f005:**
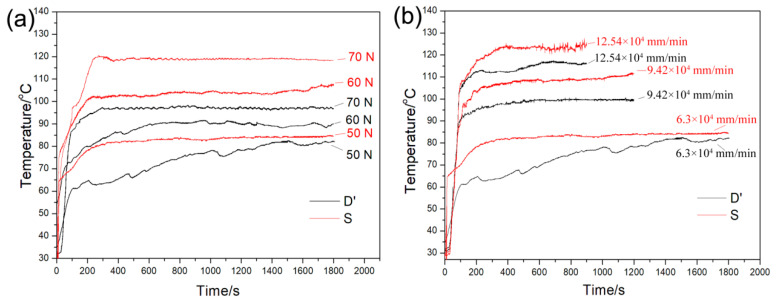
Friction temperature of the ductile iron and carbon steel wheels. (**a**) At a sliding velocity of 6.3 × 10^4^ mm/min and under different loads (**b**) with a load of 50 N and different sliding velocities and durations.

**Figure 6 materials-13-02683-f006:**
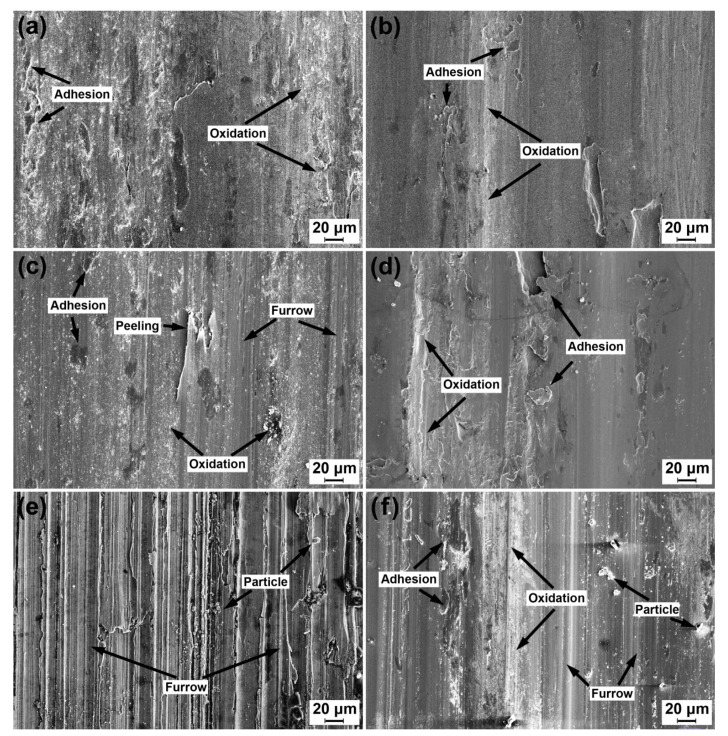
Scanning electron microscopy (SEM) micrographs of the worn surface of the ductile iron and carbon steel wheels under different loads at a sliding velocity of 6.3 × 10^4^ mm/min: (**a**) D’, 50 N, (**b**) S, 50 N, (**c**) D’, 60 N, (**d**) S, 60 N, (**e**) D’, 70 N, and (**f**) S, 70 N.

**Figure 7 materials-13-02683-f007:**
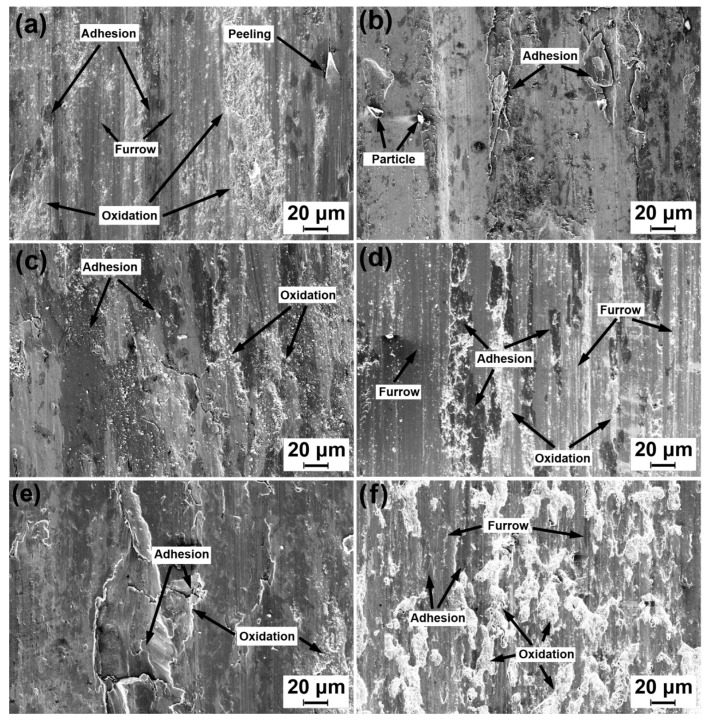
SEM micrographs of the worn surface of the ductile iron and carbon steel wheels under a normal load of 50 N at different sliding velocities: (**a**) D’, 6.3 × 10^4^ mm/min, (**b**)S, 6.3 × 10^4^ mm/min, (**c**) D’, 9.42 × 10^4^ mm/min, (**d**) S, 9.42 × 10^4^ mm/min, (**e**) D’, 12.54 × 10^4^ mm/min, and (**f**) S, 12.54 × 10^4^ mm/min.

**Figure 8 materials-13-02683-f008:**
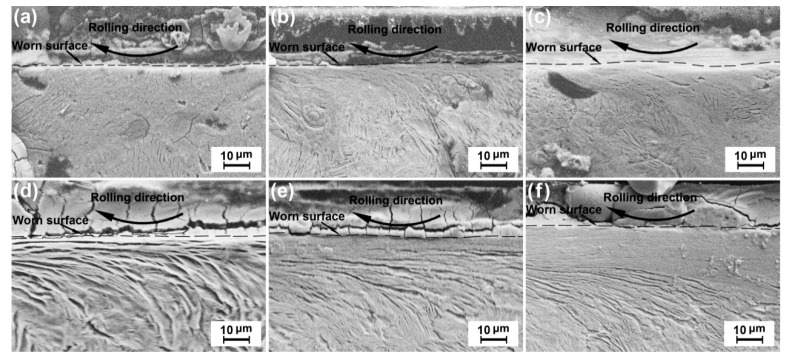
Wheel-material plastic-deformation under different loads at a sliding velocity of 6.3 × 10^4^ mm/min: (**a**) D’, 50 N, (**b**) D’, 60 N, (**c**) D’, 70 N, (**d**) S, 50 N, (**e**) S, 60 N, and (**f**) S, 70 N.

**Figure 9 materials-13-02683-f009:**
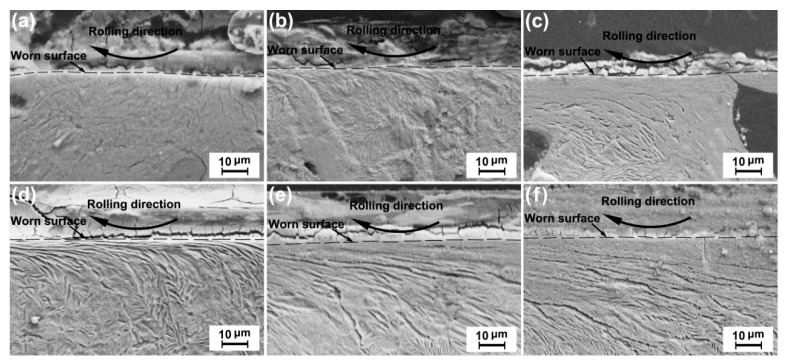
Wheel-material plastic-deformation under a normal load of 50 N at different sliding velocities: (**a**) D’ 6.3 × 10^4^ mm/min, (**b**) D’, 9.42 × 10^4^ mm/min, (**c**) D’, 12.54 × 10^4^ mm/min, (**d**) S, 6.3 × 10^4^ mm/min, (**e**) S, 9.42 × 10^4^ mm/min, and (**f**) S, 12.54 × 10^4^ mm/min.

**Figure 10 materials-13-02683-f010:**
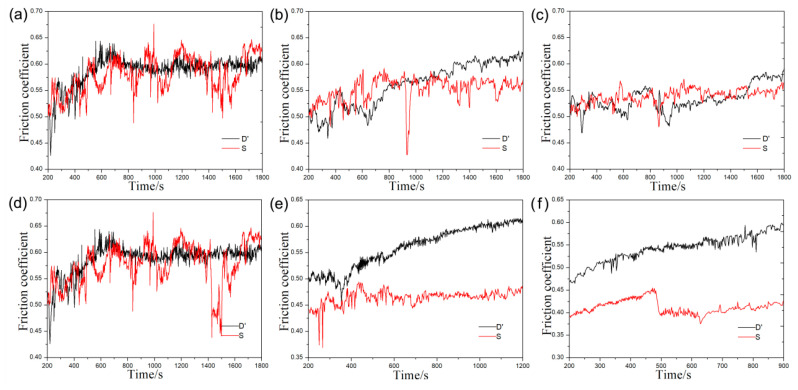
Friction-coefficient curves of the ductile iron and carbon steel wheels under different normal loads at different sliding velocities: (**a**) 50 N, (**b**) 60 N, (**c**) 70 N, (**d**) 6.3 × 10^4^ mm/min, (**e**) 9.42 × 10^4^ mm/min, and (**f**) 12.54 × 10^4^ mm/min.

**Figure 11 materials-13-02683-f011:**
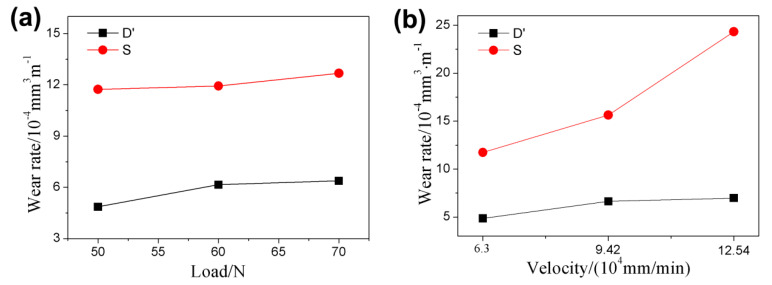
Wear rates of the ductile iron and carbon steel wheels with respect to (**a**) the normal load at 6.3 × 10^4^ mm/min and (**b**) the sliding velocity under a normal load of 50 N.

**Table 1 materials-13-02683-t001:** Chemical compositions of the testing materials (wt%).

Material	C	Mn	Si	S	P	RE	Mg	Fe
D	3.38	0.27	2.46	0.018	0.032	0.023	0.057	Bal.
S	0.61	0.72	0.27	0.004	0.008	-	-	Bal.
R	0.72	0.83	0.212	0.023	0.024	-	-	Bal.

**Table 2 materials-13-02683-t002:** Mechanical properties of the testing materials.

Material	Tensile Strength $\left(\sigma_{b}\right)$ MPa	Yield Strength $\left(\sigma_{0.2}\right)$ MPa	Elongation (A) %	Hardness HB	Impact Energy J
D	560.0	-	12.0	213	12.0
D’	917.5	798	5.0	292	4.0
S	970.7	657.3	12.5	286	32.0
R	785.0	-	12.0	240	24.0

**Table 3 materials-13-02683-t003:** Thermal diffusivity, specific heat, and thermal conductivity of the ductile iron and carbon steel wheels.

Temperature /°C	Thermal Diffusivity $/10^{-6}m^{2}\cdot s^{-1}$		Specific Heat $/J\cdot kg^{-1}K^{-1}$		Thermal Conductivity $/W\cdot m^{-1}\cdot K^{-1}$	
	S	D’	S	D’	S	D’
100	5.58	11.1	433	501	19.0	39.3
200	5.53	10.0	475	556	20.6	38.4
300	5.48	8.91	516	610	22.1	37.6
400	5.12	7.78	529	647	21.2	34.8
500	4.71	6.75	564	702	20.8	32.8

**Table 4 materials-13-02683-t004:** Sliding friction results for the ductile iron and carbon steel wheels under different loads.

Number	Normal Load /(N)	Wear Rate /(10^−4^ mm^3^m^−1^)	Friction Coefficient	Difference Value of Friction Coefficient	Wear Depth /(μm)	Grinding Crack Width /(μm)	Roughness of Worn Surface /(μm)
D’-1	50	4.86	0.59	0.076	82. 093	3347.861	16.0
D’-2	60	6.22	0.56	0.165	82. 956	3398.664	17.8
D’-3	70	6.52	0.53	0.139	129.548	3620.923	34.4
S-1	50	11.56	0.57	0.238	160.002	3773.329	30.7
S-2	60	11.93	0.55	0.223	184.877	3951.137	33.7
S-3	70	12.62	0.51	0.100	165.216	4335.328	17.9

**Table 5 materials-13-02683-t005:** Sliding-friction results for the ductile iron and carbon steel wheels with different sliding velocities and durations.

Number	Sliding Velocity /(m/s)	Wear Rate /(10^−4^ mm^3^m^−1^)	Friction Coefficient	Difference Value of Friction Coefficient	Wear Depth /(μm)	Grinding Crack Width /(μm)	Roughness of Worn Surface /(μm)
D’-4	1.05	4.86	0.59	0.076	82.993	3347.861	16.0
D’-5	1.57	6.63	0.56	0.123	132.614	3824.132	18.1
D’-6	2.09	6.98	0.54	0.158	138.698	3814.286	12.5
S-4	1.05	11.74	0.58	0.238	160.002	3773.329	33.7
S-5	1.57	15.63	0.46	0.076	212.817	4579.814	19.5
S-6	2.09	24.32	0.41	0.233	352.623	6282.893	16.0

**Table 6 materials-13-02683-t006:** Energy-dispersive X-ray spectroscopy (EDS) results of the worn surface of the ductile iron and carbon steel wheels at a sliding velocity of 6.3 × 10^4^ mm/min under different loads.

Material	C	O	Si	Fe
D’-1	11.35	10.01	2.09	76.55
D’-2	11.63	12.44	2.05	73.88
D’-3	11.76	9.67	2.08	76.48
S-1	13.98	5.48	0	80.54
S-2	7.70	32.77	0	59.53
S-3	13.00	25.46	0	61.54

**Table 7 materials-13-02683-t007:** Energy-dispersive X-ray spectroscopy (EDS) results of the worn surface of the ductile iron and carbon steel wheels under a normal load of 50 N at different sliding velocities.

Material	C	O	Si	Fe
D’-4	11.35	10.01	2.09	76.55
D’-5	11.04	13.44	1.88	73.64
D’-6	12.79	18.11	2.05	67.06
S-4	13.98	5.48	0	80.54
S-5	10.31	26.48	0	63.21
S-6	11.24	33.23	0	55.54

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
