# Peer review of "Wear Behavior of Ductile Iron Wheel Material Used for Rail-Transit Vehicles under Dry Sliding Conditions"

_materials, 2020, doi:10.3390/ma13122683_

Round 1

Reviewer 1 Report

The presented research is an interesting part of the project carried out by the authors.
It would be complemented by analyzing the cooperation of the wheel material with various materials of the braking systems.
In the future, fatigue testing should be done compared to steel wheels.
The presented part may constitute a separate scientific work. However, this partial analysis should be highlighted in the paper.
The broader context of scientific research should be described.

Reviewer 2 Report

This paper entitled "Wear Behavior of Ductile Iron Wheel Material Used for Rail-Transit Vehicles under Dry Sliding Conditions," the authors analyze the ductile iron wheel used for a rail-transit vehicle. The topic is of interest to the Materials' readers to know about the new insight of mechanical and metallurgical properties of the ductile iron wheel after heat treatment. However, some changes are needed in the paper to be considered for publication in this journal.

1) The reference format does not match the required one. It should be the surname et al., for example, Abedi et al. [17]. Please revise all the citations format.

2) It is recommended to improve the last paragraph of the introduction section. It is necessary to be more concise about the range of parameters used during sliding and enhance the novelty of this work concerning others.

3) Please revise lines 54 to 56. There is a complete duplicate sentence.

4) How are the wear test conditions (load and speed) related to the actual service conditions of this type of wheels? The speeds used seem to be very low (3.8 km/h and 7.6 km/h) respect to real scenarios.

5) It needed to mention in which direction has been done the measure the surface roughness in the worn specimens.

6) If graphite is playing the role of a lubricant, why is friction higher with nodular cast iron than with steel?

Reviewer 3 Report

Paper No.: Materials-815181

Title: Wear Behavior of Ductile Iron Wheel Material Used for Rail-Transit Vehicles under Dry Sliding Conditions

This article presents comparison of wear properties of ductile iron wheels and carbon steel wheels used in railway-transit application. Block-on-ring wear tests were performed under dry conditions at various normal load and sliding velocity combinations. Wear rate, friction coefficient as well as microstructure, mechanical and thermal properties of the two materials are compared.

The paper is well written and organized. The description of experimental procedure is detailed. The results are presented clearly and discussed in depth. There are minor grammar errors so it is recommended that author review the entire paper for grammatical errors and language clarity. Apart from that please address following concerns.

  1. The abstract can be improved. “Contrast friction test” on line 11 appears to be a typo. On line 14 and 15, please include the normal load level. Please rewrite the sentence on line 18.
  2. On line 38, please mention which industry authors are referring to.
  3. On line 54-56, the two sentences are repeated.
  4. For Figure 8, 9 and 10, please add both normal loads and sliding velocities. For example, in figure 8 sliding velocity is not mentioned.

The reviewer believes that the manuscript can be recommended for publication after minor revision after addressing the above concerns.

Reviewer 4 Report

The manuscript studies wear behaviour of railway wheel materials. Material characterization methods, including SEM-EDS, were applied and basic tribological wear tests were done. Besides typical values, such as coefficient of friction, also temperature of the contact was measured, which is an advantage. The study has direct practical application, which is an advantage of the paper as well. The manuscript is mostly well-written, and figures are clear.

Some questions and comments are presented below.

11. What is "A contrast friction test", unclear, please revise.

18. "difference value of the friction coefficient". This term "difference value" is unclear to a reader in the abstract.

43. "it was" in the middle of the sentence makes the sentence rather clumsy, please refine.

54-56. You repeat two sentences. Please delete one.

109. what do you mean by "friction distance"?

117-118. To where exactly the thermocouple was connected? How close to the contact interface? Did you measure the room temperature (and huminity)?

131. Was the surface roughnesses of the samples before testing measured? If, what was it?

197. Is the friction coefficient value in the table an average value during a test? Please define this.

197. How was wear rate determined?

197. Check the labels in the table (line breaks).

287. What is the minimum value? Is it after 200 seconds, indicated by the x-axis in Fig. 10? Why 0-200 seconds are not shown here, whereas are shown in temperature graphs?

290. Please revise the sentence "The friction is the process of node contact and slippage."

General comment. Please use consistent labels in all figures, either Figure or Fig.
